# Transfer Learning in Magnetic Resonance Brain Imaging: A Systematic Review

**DOI:** 10.3390/jimaging7040066

**Published:** 2021-04-01

**Authors:** Juan Miguel Valverde, Vandad Imani, Ali Abdollahzadeh, Riccardo De Feo, Mithilesh Prakash, Robert Ciszek, Jussi Tohka

**Affiliations:** A.I. Virtanen Institute for Molecular Sciences, University of Eastern Finland, 70150 Kuopio, Finland; juanmiguel.valverde@uef.fi (J.M.V.); vandad.imani@uef.fi (V.I.); ali.abdollahzadeh@uef.fi (A.A.); riccardo.defeo@uef.fi (R.D.F.); mithilesh.prakash@uef.fi (M.P.); robert.ciszek@uef.fi (R.C.)

**Keywords:** transfer learning, magnetic resonance imaging, brain, systematic review, survey, machine learning, artificial intelligence, convolutional neural networks

## Abstract

(1) Background: Transfer learning refers to machine learning techniques that focus on acquiring knowledge from related tasks to improve generalization in the tasks of interest. In magnetic resonance imaging (MRI), transfer learning is important for developing strategies that address the variation in MR images from different imaging protocols or scanners. Additionally, transfer learning is beneficial for reutilizing machine learning models that were trained to solve different (but related) tasks to the task of interest. The aim of this review is to identify research directions, gaps in knowledge, applications, and widely used strategies among the transfer learning approaches applied in MR brain imaging; (2) Methods: We performed a systematic literature search for articles that applied transfer learning to MR brain imaging tasks. We screened 433 studies for their relevance, and we categorized and extracted relevant information, including task type, application, availability of labels, and machine learning methods. Furthermore, we closely examined brain MRI-specific transfer learning approaches and other methods that tackled issues relevant to medical imaging, including privacy, unseen target domains, and unlabeled data; (3) Results: We found 129 articles that applied transfer learning to MR brain imaging tasks. The most frequent applications were dementia-related classification tasks and brain tumor segmentation. The majority of articles utilized transfer learning techniques based on convolutional neural networks (CNNs). Only a few approaches utilized clearly brain MRI-specific methodology, and considered privacy issues, unseen target domains, or unlabeled data. We proposed a new categorization to group specific, widely-used approaches such as pretraining and fine-tuning CNNs; (4) Discussion: There is increasing interest in transfer learning for brain MRI. Well-known public datasets have clearly contributed to the popularity of Alzheimer’s diagnostics/prognostics and tumor segmentation as applications. Likewise, the availability of pretrained CNNs has promoted their utilization. Finally, the majority of the surveyed studies did not examine in detail the interpretation of their strategies after applying transfer learning, and did not compare their approach with other transfer learning approaches.

## 1. Introduction

Magnetic resonance imaging (MRI) is an non-invasive imaging technology that produces three dimensional images of living tissue. MRI measures radio-frequency signals emitted from hydrogen atoms after the application of electromagnetic (radio-frequency) waves, localizing the signal using spatially varying magnetic gradients, and is capable to measure various properties of the tissue depending on the particular pulse sequence applied for the measurement [1]. Particularly, anatomical (or structural) MRI is used to generate images of the anatomy of the studied region. Functional MRI (fMRI) demonstrates regional, time-varying changes in brain metabolism in the form of increased local cerebral blood flow (CBF) or in the form of changes in oxygenation concentration (Blood Oxygen Level Dependent, or BOLD contrast) that are both consequences of increased neural activity [2]. Diffusion MRI (dMRI) is used to assess the microstructural properties of tissue based on the diffusive motion of water molecules within each voxel [3]. The use of MRI is increasing rapidly, not only for clinical purposes but also for brain research and development of drugs and treatments. This has called for machine learning (ML) algorithms for automating the steps necessary for the analysis of these images. Common tasks for machine learning include tumor segmentation [4], registration [5], and diagnostics/prognostics [6].

However, the variability in, for instance, image resolution, contrast, signal-to-noise ratio or acquisition hardware leads to distributional differences that limit the applicability of ML algorithms in research and clinical settings alike [7,8]. In other words, an ML model trained for a task in one dataset may not necessarily be applied to the same task in another dataset because of the distributional differences. This difficulty emerges in large datasets that combine MR images from multiple studies and acquisition centers since different imaging protocols and scanner hardware are used, and also hampers the clinical applicability of ML techniques as the algorithms would need to be re-trained in a new environment. Additionally, partly because of the versatility of MRI, an ML model trained for a task could be useful in another, related task. For instance, an ML model trained for brain extraction (skull-stripping) might be useful when training an ML model for tumor segmentation. Therefore, developing strategies to address the variation in data distributions within large heterogeneous datasets is important. This review focuses on transfer learning, a strategy in ML to produce a model for a target task by leveraging the knowledge acquired from a different but related source domain [9].

Transfer learning (or knowledge transfer) reutilizes knowledge from source problems to solve target tasks. This strategy, inspired by psychology [10], aims to exploit common features between related tasks and domains. For instance, an MRI expert can specialize in computed tomography (CT) imaging faster than someone with no knowledge in either MRI or CT. There exist several surveys [9,11,12] of transfer learning in machine learning literature, including in medical imaging [13], but to our knowledge, no surveys have focused on its use in brain imaging or MRI applications. Pan and Yang [11] presented one of the earliest surveys in transfer learning, which focused on tasks with the same feature space (i.e., homogeneous transfer learning). Pan and Yang [11] proposed two transfer learning categorizations that are in wide use. One categorization divided approaches based on the availability of labels, and the other based on which knowledge is transferred (e.g., features, parameters). Day and Khoshgoftaar [9] surveyed heterogeneous transfer learning applications, i.e., tasks with different feature spaces, and, also based on the availability of labels, transfer learning approaches were divided into 38 categories. More recently, another survey [12] expanded the number of categories to over 40. These recent categorizations can be useful in a general context as they were derived from approaches from diverse areas. However, the large number of proposed categories can lead to their underutilization and the classification of similar strategies differently in specific fields, such as MR brain imaging. We chose the categorization proposed by Pan and Yang [11] since it divides transfer learning approaches into few categories, and because such categorization is the basis of categorization in other transfer learning surveys [9,12,13]. Furthermore, we introduced new subcategories that describe how transfer learning was applied, revealing widely used strategies within the MR brain imaging community.

Systematic reviews analyze methodologically articles from specific areas of science. Recent systematic reviews in biomedical applications of machine learning have covered such topics as predicting stroke [14], detection and classification of transposable elements [15], and infant pain prediction [16]. Here, we present the first systematic review of transfer learning in MR brain imaging applications. The aim of this review is to identify research trends, and to find popular strategies and methods specifically designed for brain applications. We highlight brain-specific methods and strategies addressing data privacy, unseen target domains, and unlabeled data—topics especially relevant in medical imaging. Finally, we discuss the research directions and knowledge gaps we found in the literature, and suggest certain practices that can enhance methods’ interpretability.

## 2. Transfer Learning

According to [11], we define a domain in transfer learning as D={X,P(X)}, where X is the feature space, and P(X) with $X=\left\{x_{1},\ldots ,x_{n}\right\}\subset X$ is a marginal probability distribution. For example, X could include all possible images derived from a particular MRI protocol, acquisition parameters, and scanner hardware, and P(X) depend on, for instance, subject groups, such as adolescents or elderly people. Tasks comprise a label space Y and a decision function *f*, i.e., T={Y,f}. The decision function is to be learned from the training data (X,Y). Tasks in MR brain imaging can be, for instance, survival rate prediction of cancer patients, where *f* is the function that predicts the survival rate, and Y is the set of all possible outcomes. Given a source domain $D_{S}$ and task $T_{S}$, and a target domain $D_{T}$ and task $T_{T}$, transfer learning reutilizes the knowledge acquired in $D_{S}$ and $T_{S}$ to improve the generalization of $f_{T}$ in $D_{T}$ [11]. Importantly, $D_{S}$ must be related to $D_{T}$, and $T_{S}$ must be related to $T_{T}$ [17]; otherwise, transfer learning can worsen the accuracy on the target domain. This phenomenon, called negative transfer, has been recently formalized in [18] and studied in the context of MR brain imaging [19].

Transfer learning approaches can be categorized based on the availability of labels in source and/or target domains during the optimization [11]: unsupervised transfer learning (unlabeled data), transductive (labels available only in the source domain), and inductive approaches (labels available in the target domains and, optionally, in the source domains). Table 1 illustrates these three types with examples in MR brain imaging applications.

Following the simple categorization described in Pan and Yang [11], transfer learning approaches can be grouped into four categories based on the knowledge transferred. Instance-based approaches estimate and assign weights to images to balance their importance during optimization. Feature-based approaches seek a shared feature space across tasks and/or domains. These approaches can be further divided into asymmetric (transforming target domain features into the source domain feature space), and symmetric (finding a common intermediate feature representation). Parameter-based approaches find shared priors or parameters between source and target tasks/domains. Parameter-based approaches assume that such parameters or priors share functionality and are compatible across domains, such as a domain-invariant image border detector. Finally, relational-based approaches aim to exploit common knowledge across relational domains.

### Related Approaches

There exist various strategies to improve ML generalization in addition to transfer learning. In multi-task learning, ML algorithms are optimized for multiple tasks simultaneously. For instance, Weninger et al. [20] proposed a multi-task autoencoder-like convolutional neural network with three decoders—one per task—to segment and reconstruct brain MR images containing tumors. Zhou et al. [21] presented an autoencoder with three branches for coarse, refined, and detailed tumor segmentation. Multi-task learning differs from transfer learning in that transfer learning focuses on target tasks/domains whereas multi-task learning tackles multiple tasks/domains simultaneously; thus, considering source and target tasks/domains equally important [11]. Data augmentation, which adds slightly transformed copies of the training data to the training set, can also enhance algorithms’ extrapolability [22,23]. Data augmentation is useful when images with certain properties (i.e., a specific contrast) are scarce, and it can complement other techniques, such as multi-task or transfer learning. However, in contrast to multi-task or transfer learning, data augmentation ignores whether the knowledge acquired from similar tasks can be reutilized. Additionally, increasing the training data also increases computational costs, and finding advantageous data transformations is non-trivial. Finally, meta-learning aims to find parameters or hyper-parameters of machine learning models to generalize well across different tasks. In contrast to transfer learning, meta-learning does not focus on a specific target task or domain, but on all possible domains, including unseen domains. Liu et al. [24] showed that the model-agnostic meta-learning strategy [25] yields state-of-the-art segmentations in MR prostate images.

## 3. Methods

We followed the PRISMA statement [26]. The PRISMA checklist is included in the Appendix A.

### 3.1. Search Strategy

We searched articles about transfer learning applied to MR images in the Scopus database (https://www.scopus.com, accessed on 19 October 2020). In addition, we searched for relevant articles from the most recent (2020) Medical Image Computing and Computer Assisted Interventions (MICCAI) conference from the Springer website (https://link.springer.com/search?facet-conf-event-id=miccai2020&facet-content-type=Chapter, accessed on 30 October 2020) because they were unavailable in Scopus at the time when the search was performed. To ensure we retrieved relevant results, we focused on finding certain keywords in either the title, abstract, or article keywords. We also searched for “knowledge transfer” and "domain adaptation" as these are alternative names or subclasses of transfer learning. Similarly, we searched for different terms related to MRI, including magnetic resonance and diffusion imaging. Table 2 shows the exact searched terms used. Note that “brain” was not one of the keywords as we observed that including it would have led to the omission of several relevant articles.

### 3.2. Study Selection

We excluded non-article records retrieved (e.g., entire conferences proceedings). We reviewed the abstracts of all candidate articles to discard studies that were not about MR brain imaging or did not apply transfer learning. For this, each abstract was reviewed by two co-authors. In more detail, we used the following procedure to assign the abstracts to co-authors. JMV assigned the abstracts randomly to six labels so to that each label corresponded to one co-author (all co-authors except JMV). JT assigned the labels with co-authors so that JMV was unaware of the true identity of each reviewer to guarantee anonymity and to reduce reviewer bias during the screening. Furthermore, to ensure the same criteria were applied during this review by each co-author, review guidelines written by JT and JVM and commented by other co-authors were distributed among the co-authors. Reviewers determined the relevance of each abstract they reviewed and, to support their decision, reviewers also included extra information from the screened abstracts, including the studied organs (e.g., brain, heart), MRI modality (e.g., anatomical, functional), and comments. If the two reviewers disagreed about the relevance of an abstract, JVM solved the disagreement.

### 3.3. Data Collection

Articles that were deemed relevant based on their abstracts were randomly and evenly divided among all seven co-authors for their full review (based on the complete article). We extracted information to categorize the surveyed papers, including the methods, datasets, tasks, types of input features, and whether the labels were available in source and/or in target domain. To ensure criteria homogeneity in the classification, JVM verified the collected data and discussed disagreements with the other co-authors. The information extracted in this process is in the Appendix A.

We discarded studies that were irrelevant to this survey (i.e., not about transfer learning or MR brain imaging), too unclear to obtain relevant information, did not perform a proper experimental evaluation of the transfer learning approach, or we could not access. The authors of articles that we could not access were additionally contacted through https://www.researchgate.net/ (accessed on 4 December 2020) to obtain the article. Additionally, we included and examined other relevant articles coming to our attention during the review of the articles.

### 3.4. Screening Process

Figure 1 summarizes the screening process and the number of articles after each step. First, we retrieved 399 records from Scopus (19 October 2020) and 34 from Springer (30 October 2020). We excluded 42 records from Scopus: 41 entire proceedings of conferences and one corrigendum, leaving 391 journal and conference articles. After the abstract review, we excluded 220 articles that were either not about transfer learning or MR brain imaging. We reviewed the remaining 171 articles based on full paper, and 44 articles were discarded: 26 studies mixed data from the same subject to their training and testing sets, 8 were unrelated to transfer learning in MR brain imaging, 7 were unclear, 2 were inaccessible, and 1 was a poster. The studies that mixed data from the same subject typically performed ML tasks on slices of MRI instead of MRI volumes, and both training and testing sets appeared to contain data from the same volumetric MRI. Finally, we added two relevant articles coming to our attention while reading the articles, resulting in 129 articles that were included in this survey.

## 4. Results

### 4.1. Applications

We found 29 applications that we considered distinct. Table 3 summarizes the distinct tasks and applications within these tasks. Note that some articles studied more than one task/application. Figure 2 (left) shows the number of articles that addressed different brain diseases according to the 11th International Classification of Diseases (ICD-11) (https://icd.who.int/en, accessed on 18 November 2020).

Classification tasks were the most widely studied and, among these, dementia-related (neurocognitive impairment, Figure 2 (left)) and tumor-related (neoplasms, Figure 2) applications accounted for 45.59% and 14.71% of all classification tasks, respectively. Other applications included autism spectrum disorder diagnostics, and functional MRI decoding that is classification of, for instance, stimulus, or cognitive state based on observed fMRI [40]. Segmentation was the second most popular task, studied in one-third of the articles. Segmentation applications included anatomical, lesion, and tumor segmentation with each application studied in approximately one-third of the segmentation articles. Regression problems were considerably less common than classification and segmentation, and, within these, age prediction was predominant. Among the other applications, image reconstruction and registration were the most widely studied.

Figure 2 (right) shows the number of articles that employed anatomical, functional, diffusion MRI, and multimodal data. The majority of the surveyed articles (98) utilized anatomical MR images, including T1-weighted, T2-weighted, FLAIR, and other contrasts. The number of studies that utilized fMRI and diffusion MRI data was 18 and 6, respectively. Finally, 8 studies were multimodal, combining MRI with positron emission tomography (PET) or CT.

### 4.2. Machine Learning Approaches

Figure 3 (left) illustrates the number of articles that utilized specific machine learning and statistical methods. Convolutional neural networks (CNNs) were applied in the majority of articles (68.22%), followed by kernel methods (including support vector machines (SVMs) and support vector regression), multilayer perceptrons, decision trees (including random forests), Bayesian methods, clustering methods, elastic net, long short-term memory networks, and deep neural networks without convolution layers. Several other methods (Figure 3 (left), "Others") appeared only in a single article: deformable models, manifold alignment, graph neural networks, Fisher’s linear discriminant, principal component analysis, independent component analysis, joint distribution alignment, singular value decomposition, Pearson correlation, and Adaboost. Figure 3 (right) shows the number of articles that utilized CNNs, kernel methods, and other approaches across years. Since 2014, the number of articles applying transfer learning to MR brain imaging applications and the use of CNNs have grown exponentially. In the last two years 80% of the articles (68) utilized CNNs, and, during this period, their use and the total number of articles have started to converge. Finally, we computed a contingency table between the machine learning methods and the brain disease categories in the surveyed papers, but found no correlation between methods and disease categories (see Appendix A).

### 4.3. Transfer Learning Approaches

Table 4 summarizes the transfer learning strategies found in the surveyed papers. Note that certain articles applied multiple strategies. We classified these strategies into instance, feature representation ("feature" for short), parameter, and relational knowledge [11]. Afterwards, we divided feature-based approaches into symmetric and asymmetric. Since the categorization described in [11] is general, we propose new subcategories, described below. These new subcategories, based on the strategies found during our survey, aim to reduce ambiguity and to facilitate the identification of popular transfer learning methods in the context of MR brain imaging.

We divided instance-based approaches into fixed and optimized sub-categories. Fixed weights are those weights assigned following certain preset criteria, such as assigning higher weights to target domain images. On the other hand, optimized weights are weights estimated by solving an optimization problem. Furthermore, we separated optimized weights approaches based on whether such optimization problem required labels—requirement not always feasible in medical imaging—into supervised and unsupervised.

We divided symmetric feature-based approaches that find a common feature space between source and target domain were into direct and indirect subcategories. Direct approaches are methods that operate directly on the feature representations of source and target domain data, thereby requiring these data simultaneously. We considered such requirement an important discriminative since it can limit the applicability of the approach. More precisely, approaches that require source and target domain data simultaneously may need more memory and, importantly, may not be applicable if source and target domain data cannot be shared due to privacy issues. As an example of this category, consider an approach that minimizes the distance between the feature representations of source and target domain images, aiming to transform the data into the same feature space. In contrast, indirect approaches do not operate directly on the feature representations and do not need simultaneous access to source and target domain data.

Parameter-based approaches consisted of two steps: first, finding the shared priors or parameters, and second, fine-tuning all or certain parameters in the target domain data. Since the approaches that fine-tuned all parameters assumed that their previous parameters were closer to the solution than their random initialization, we considered these approaches as prior sharing. Although sharing parameters could be also considered as sharing priors, separating these two approaches based on whether certain or all model parameters were fine-tuned revealed the popularity of different strategies. Furthermore, we propose to divide parameter-sharing approaches into two categories: approaches that only utilize one model, and approaches in which the shared parameters correspond to a feature-extracting model for optimizing separate models, thereby comprising multiple models.

### 4.3.1. Instance-Based Approaches

We found 16 approaches (11.63% of the studies) that applied transfer learning by weighing images. We propose to divide these approaches based on whether images’ weights were fixed or optimized, and in the latter case, distinguish between supervised or unsupervised optimization.

Fixed-weights strategies included sample selection (i.e., binary weights), such as in [41], where authors discarded certain images in datasets biased with more subjects with a given pathology. Likewise, Cheng et al. [42,43] trained a classifier to perform sample selection based on the images’ probability these images belong to the source domain. Similar probabilities and non-binary fixed weights were applied in [44,45,46], respectively, allowing the contribution of all images in the studied task.

Images’ weights that were optimized via unsupervised strategies relied on data probability density functions (PDFs). The surveyed articles that applied these strategies optimized images’ weights separately before tackling the studied task (e.g., Alzheimer’s diagnostics/prognostics). Images’ weights were derived by minimizing the distance (Kullback-Leibler divergence, Bhattacharyya, squared Euclidean, and maximum mean discrepancy) between the PDFs in the source and target domains [47,48,49,50]. On the other hand, supervised strategies optimized images’ weights and the ML model simultaneously by minimizing the corresponding task-specific loss [45,51,52,53,54]. Notably, supervised and unsupervised strategies can be combined, and approaches can also incorporate extra information unused in the main task. For instance, Wachinger et al. [55] also considered age and sex during the unsupervised optimization of PDFs for Alzheimer’s diagnostics/prognostics.

### 4.3.2. Feature-Based Approaches

We found 38 approaches (29.46% of the studies) that applied transfer learning by finding a common feature space between source and target domains/tasks. These approaches were divided into symmetric and asymmetric (see Section 2). Additionally, we propose to subdivide symmetric approaches based on whether the common feature space was achieved by directly operating on the source and target feature representations (e.g., by minimizing their distance), or indirectly.

We found 7 asymmetric approaches that transformed target domain features into source domain features via generative adversarial networks [56,57], Bregman divergence minimization [58], probabilistic models [59], median [60], and nearest neighbors [61]. Contrarily, Qin et al. [62] transformed source domain features into target domain features via dictionary-based interpolation to optimize a model on the target domain.

Among the surveyed 31 symmetric approaches, direct approaches operated on the feature representations across domains by minimizing their differences (via mutual information [63], maximum mean discrepancy [46,49,64], Euclidean distance [65,66,67,68,69,70,71], Wasserstein distance [72], and average likelihood [73]), maximizing their correlation [74,75] or covariance [36], and introducing sparsity with L1/L2 norms [42,76]. On the other hand, indirect approaches were applied via adversarial training [28,41,54,77,78,79,80,81,82,83,84,85], and knowledge distillation [86].

### 4.3.3. Parameter-Based Approaches

We found 87 approaches (65.89% of the studies) that applied transfer learning by sharing priors or parameters. Parameter-sharing approaches were further subdivided based on whether one model or multiple models were optimized.

The most common approach to apply a prior-sharing strategy—and, in general, transfer learning—was fine-tuning all the parameters of a pretrained CNN [29,31,32,33,35,39,71,87,88,89,90,91,92,93,94,95,96,97,98,99,100,101,102,103,104,105,106,107,108,109,110,111,112,113,114,115,116,117,118,119] (80% of all prior-sharing methods). Other approaches utilized Bayesian graphical models [37,38,120,121], graph neural networks [122], kernel methods [64,123], multilayer perceptrons [124], and Pearson-correlation methods [125]. Additionally, Sato et al. [27] proposed a general framework to inhibit negative transfer. Within the prior-sharing group, 20 approaches utilized a parameter initialization derived from pretraining on natural images (i.e., ImageNet [126]) whereas 26 approaches pretrained on medical images.

The second most popular strategy to apply transfer learning was fine-tuning certain parameters in a pretrained CNN [34,127,128,129,130,131,132,133,134,135,136,137,138,139,140,141,142,143,144,145,146]. The remaining approaches first optimized a feature extractor (typically a CNN or a SVM), and then trained a separated model (SVMs [30,45,147,148,149], long short-term memory networks [150,151], clustering methods [148,152], random forests [70,153], multilayer perceptrons [154], logistic regression [148], elastic net [155], CNNs [156]). Additionally, Yang et al. [157] ensembled CNNs and fine-tuned their individual contribution. Within the parameter-sharing group, 17 approaches utilized a ImageNet-pretrained CNN, and 15 others pretrained on medical images.

We found 40 studies that utilized publicly-available CNN architectures. The most popular were VGG [158] (23), ResNet [159] (15), and Inception [160,161,162] (11). Nearly three-fourths of these studies fine-tuned the ImageNet-pretrained version of the CNNs whereas one-third pretrained the networks in medical datasets. Finally, among these 40 studies, 13 articles compared the performance of multiple architectures, and VGG (4) and Inception (4) usually provided the highest accuracy.

### 4.4. Transfer Learning Approaches Relevant to MR Brain Imaging

We closely examined transfer learning approaches that were inherently unique to brain imaging. Additionally, we included strategies that considered data privacy, unseen target domains, and unlabeled data, as these topics are especially relevant to the medical domain.

Brain MRI-specificity

Aside from employing brain MRI-specific input features (e.g., brain connectivity matrices in fMRI, diffusion values in dMRI) or pre-processing (e.g., skull-stripping), we only found two transfer learning strategies unique to brain imaging. Moradi et al. [36] considered cortical thickness measures of each neuroanatomical structure separately, yielding multiple domain-invariant feature spaces—one per brain region. This approach diverges from the other surveyed feature-based strategies that sought a single domain-invariant feature space for all the available input features. Cheng et al. [42] extracted the anatomical volumes of 93 regions of interest in the gray matter tissue of MRI and PET images. Afterwards, authors applied a feature-based transfer learning approach with sparse logistic regression separately in MRI and PET images, yielding informative brain regions for each modality. Finally, authors combined these features with cerebrospinal fluid (CSF) biomarkers, and applied an instance-based approach to find informative subjects.

Privacy

We found two frameworks that considered privacy. Li et al. [54] built a federated-learning framework that optimized a global model by transferring the parameters of site-specific local models. This framework requires no database sharing, and the transferred parameters are slightly perturbed to ensure differential privacy [163]. Sato et al. [27] proposed a framework that applies online transfer learning and only requires the output—not the images—of the models optimized in the source domain data. Notably, this framework also tackles negative transfer. Besides these two frameworks, parameter-based approaches that were fine-tuned exclusively in target domain data also protected subjects’ privacy since they required no access to source domain images during the fine-tuning.

Unseen Target Domains

We found four studies that considered unseen target domains. van Tulder and de Bruijne [70], Shen and Gao [78] utilized multimodal images and proposed a symmetric feature-based approach that drops a random image modality during the optimization, avoiding the specialization to any specific domain. Hu et al. [38], Varsavsky et al. [84] considered that each image belongs to a different target domain, and presented a strategy that adapts to each image. Note that these approaches also protected data privacy as no access to source domain images was required to adapt to the target domain.

Unlabeled Data

We revisited transfer learning strategies that handled unlabeled data in the target domain and, optionally, in the source domain. Van Opbroek et al. [47] presented an instance-based approach that relied exclusively on the difference between data distributions, thereby requiring no labels. Goetz et al. [44] expanded this idea and also incorporated source domain labels to optimize a model for estimating images’ weights. On the other hand, feature-based approaches can include source domain labels while finding appropriate feature transformations. Following this idea, Li et al. [58] minimized the Bregman distance between the source and target domain features while optimizing a task-specific classifier. Similarly, Ackaouy et al. [72], Orbes-Arteaga et al. [82] sought a shared feature space while minimizing Dice loss and a consistency loss, respectively, with source domain labels. Additionally, various indirect symmetric feature-based approaches jointly optimized an adversarial loss and a task-specific loss on the source domain images [79,80,83]. Finally, Orbes-Arteainst et al. [86] used knowledge distillation, training a teacher model on the labeled source domain, and optimizing a student network on the probabilistic maps from the teacher model derived with the source and target domain images.

### 4.5. Tackling Transfer Learning Issues

The application of transfer learning has a few potential detrimental consequences that only two studies included in the survey have investigated. Kollia et al. [136] considered source and target domain images jointly during the optimization to avoid catastrophic forgetting—lower performance on the source domain after applying transfer learning to the target domain. Sato et al. [27] proposed an algorithm to detect aneurysms that directly inhibits negative transfer [18]—worse results on the target domain than if no transfer learning was applied.

## 5. Discussion

### 5.1. Research Directions of Transfer Learning in Brain MRI

We surveyed 129 articles on transfer learning in brain MRI, and our results indicate an increased interest in the field in recent years. Alzheimer’s diagnostics/prognostics, tumor classification, and tumor segmentation were the most studied applications (see Figure 2 (left), and Table 3). The popularity of these applications is likely linked to the existence of well-known publicly or easily available databases, such as ADNI [164], and the Brain Tumor Segmentation challenge (BraTS) datasets [4,165,166]. We would like to point out that there are also other large MRI databases available to researchers (e.g., ABIDE [167,168], Human Connectome Project [169]), but the number of articles in this review utilizing these other databases was considerably lower than ADNI or BraTS. CNNs were the most used machine learning method, utilized in 68.22% of all the reviewed papers, and 80% in the last two years (Figure 3). The demonstrations of outperformance of CNNs over other methods [170], and the availability of trained CNNs in ImageNet [126], such as AlexNet [171], VGG [158], and Inception [160], probably explains CNNs’ popularity.

We classified transfer learning approaches into instance, feature representation, parameter, and relational knowledge [11]. We noticed that this classification was too coarse, hindering the identification of popular solutions within the MR brain imaging community. As Pan and Yang [11] indicated, this categorization is based on "what to transfer" rather than "how to transfer." Therefore, based on the surveyed articles, we refined this categorization by introducing subcategories that define transfer learning approaches more precisely (see Section 4.3 and Table 4). Our categorization divided instance-based approaches based on whether images’ weights were fixed or optimized, and in the latter case, subdivided to separate supervised and unsupervised optimization. Symmetric feature-based approaches were split depending on whether the strategy operated directly or indirectly on the feature representations between domains. Parameter-sharing-based approaches were divided based on whether one or multiple models were optimized. With our categorization, we found that most of the strategies pretrained a model or utilized a pretrained model, and subsequently fine-tuned all, certain parameters, or a separate model on the target domain data. Among these studies, a similar number of approaches either utilized ImageNet-pretrained CNNs or pretrained on medical images.

### 5.2. Knowledge Gaps

Beyond showing accuracy gains, the surveyed articles rarely examined other approach-specific details. Only a few studies that optimized images’ weights [49,54,55] showed their post-optimization distribution. Interestingly, several of these weights became zero or close to zero, indicating that the contribution of their corresponding images to tackle the studied task was negligible. The incorporation of sparsity-inducing regularizers as in [42], and a closer view to these weights could lead to intelligent sample selection strategies and advances in curriculum learning [172]. Regarding feature-based transfer learning approaches, various studies [28,41,46,54,58,66,70,71,76] illustrated, typically with t-SNE [173], that the source and target domain images lied closer in the feature space after the application of their method. However, we found no articles that compared and illustrated the feature space after implementing different strategies. This also raises the question of how to properly quantify that a feature space distribution is better than another in the context of transfer learning.

With a limited number of training data in the target domain, fine-tuning ImageNet-pretrained CNNs has been demonstrated to yield better results than training such CNNs from scratch, even in the medical domain [174]. In agreement with this observation, nearly half of the parameter-based approaches followed this practice. Fine-tuning all parameters (prior-sharing) and fine-tuning certain parameters (parameter-sharing) were widely used methods, although in the latter case we rarely found justifications for choosing which parameters to fine-tune. Since the first layers of CNNs capture low-level information, such as borders and corners, various studies [34,134,135,137,145] have considered that those parameters can be shared across domains. Besides, as adapting pretrained CNNs to the target domain data requires, at least, replacing the last layer of these models, researchers have likely turn fine-tuning only this randomly-initialized layer into common practice, although we found no empirical studies that supported such practice. Four surveyed articles studied different fine-tuning strategies with CNNs pretrained on ImageNet [96,134] and medical images [129,130]. The approaches that utilized ImageNet-pretrained CNNs [96,134] reported that fine-tuning more layers led to higher accuracy, suggesting that the first layers of ImageNet-pretrained networks—that detect low-level image characteristics, such as corners and borders—may not be adequate for medical images. Furthermore, Bashyam et al. [101] reported that Inceptionv2 [161] pretrained on medical images outperformed its ImageNet-pretrained version in Alzheimer’s, mild cognitive impairment, and schizophrenia classification. Finally, a recent study [130] showed that fine-tuning the first layers of a CNN yielded better results than the traditional approach of exclusively fine-tuning the last layers, suggesting that the first layers encapsulate more domain-dependent information.

The number and which parameters require fine-tuning could depend on the target domain/task and on the training set size, as large models require more training data. Likewise, the training set size in the target domain could be task-dependent. Only a few studies investigated the application of transfer learning with different training set sizes [99,101,128,129,130,135,147,150]. Among these, most articles reported that with a sufficiently large training set, models trained from scratch achieved similar or even better [99,135] results than applying transfer learning.

### 5.3. Limitations

A limitation of this survey arises from the subjectivity and difficulty of classifying transfer learning methods into the sub-categories. Additionally, a few surveyed articles combined multiple transfer learning approaches, hindering their identification and classification. Occasionally, determining the source and target tasks/domains and whether labels were used exclusively in the transfer learning approach was challenging. Thus, the numbers of approaches belonging to specific categories might not be exact. Another limitation is that, especially with CNNs, it is sometimes ambiguous to decide whether an approach is a transfer learning approach. We based our literature search largely on the authors’ opinions on whether their article was about transfer learning. Additionally, when rejecting articles from review, we typically trusted authors’ judgments: we accepted if they indicated that their approach was transfer learning. Thus, borderline articles, which the authors themselves did not categorize as transfer learning, might have escaped our literature search, despite similar approaches possibly being included in this review.

## 6. Conclusions

The growing number of transfer learning approaches for brain MRI applications signals the perceived importance of this field. The need for transfer learning arises from the heterogeneity in large datasets that combine MR images from multiple studies and acquisition centers with different imaging protocols and scanner hardware. Additionally, transfer learning can increase ML clinical applicability by simplifying the training in new environments, hence the increased interest in transfer learning.

Well-known datasets that are easily available to researchers have clearly boosted certain applications, such as Alzheimer’s diagnostics/prognostics and tumor segmentation. Similarly, the availability of pretrained CNNs has contributed to their popularization in transfer learning. Indeed, we found that pretraining a CNN, or utilizing a pretrained CNN, to fine-tune it on target domain data was the most widely used approach to apply transfer learning. Additionally, we noticed that the studies that investigated different fine-tuning strategies in ImageNet-pretrained CNNs reported higher accuracy after fine-tuning all the parameters and not just the last layers, as many other approaches did.

We found various studies tackling issues relevant to the medical imaging community, such as privacy, although we only found two brain-specific approaches that, coincidentally, did not utilize CNNs. Besides, the surveyed studies rarely examined in depth their solutions after applying transfer learning. For example, instance-based approaches seldom interpreted images’ weights; feature-based approaches did not compare various feature spaces derived with other methods; and the majority of parameter-based approaches relied on assumptions to decide which parameters to share.

The importance of enhancing an algorithm’s capability to generalize large and heterogeneous datasets will likely continue to boost transfer learning in MR brain imaging. Particularly, the need for such large databases will probably motivate the collaborations among research institutions and the development of privacy-preserving methods that avoid data sharing [54]. Transfer learning in MR brain imaging can benefit from domain expertise for investigating brain-specific approaches, such as [36,42]. Therefore, we believe that the current lack of brain imaging-specific methods will motivate the development of such methods. Additionally, the number of recent studies that investigated which CNN layers to fine-tune suggests that there will be more effort put into demonstrating and understanding when and which parameters can be shared across domains.

## Figures and Tables

**Figure 1 jimaging-07-00066-f001:**
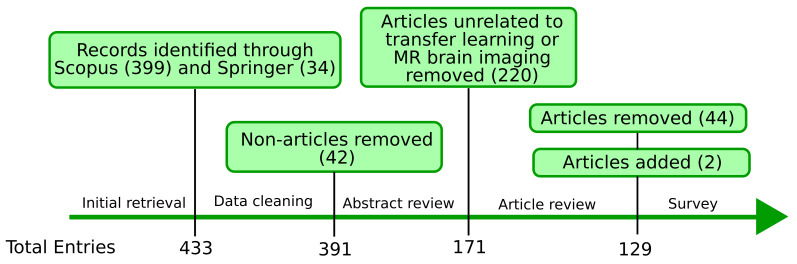
Flowchart of the screening process.

**Figure 2 jimaging-07-00066-f002:**
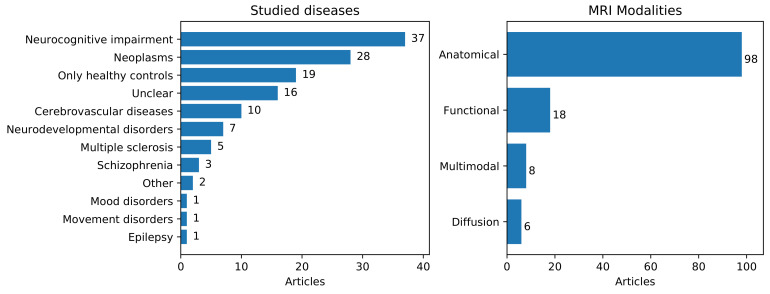
(**Left**): Number of articles according to the ICD-11 category. (**Right**): Number of articles per MRI modality.

**Figure 3 jimaging-07-00066-f003:**
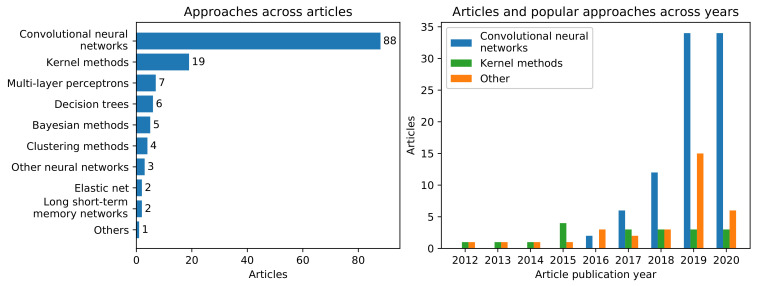
(**Left**): Number of articles according to the ML method studied (methods enumerated in Section 4.2). (**Right**): Distribution of articles according to the publication year.

**Table 1 jimaging-07-00066-t001:** Types of transfer learning and examples in MR brain imaging. ∼ indicates “different but related”. The subscripts *S* and *T* indicate source and target, respectively.

Type	Properties	Approach Example
Unsupervised	$D_{S}∼D_{T},T_{S}=T_{T}$	Transforming T1- and T2-weighted images into the same feature space with adversarial training.
Transductive	$D_{S}∼D_{T},T_{S}=T_{T}$	Learning a feature mapping from T1- to T2-weighted images while optimizing to segment tumors in T2-weighted images.
Inductive	$D_{S}∼D_{T},T_{S}∼T_{T}$	Optimizing a classifier on a natural images dataset, and fine-tuning certain parameters for tumor segmentation.
	$D_{S}∼D_{T},T_{S}=T_{T}$	Optimizing a lesion segmentation algorithm in T2-weighted images, and re-optimizing certain parameters on FLAIR images.
	$D_{S}=D_{T},T_{S}∼T_{T}$	Optimizing a lesion segmentation algorithm in T2-weighted images, and re-optimizing certain parameters in the same images for anatomical segmentation.

**Table 2 jimaging-07-00066-t002:** Search terms in Scopus and Springer.

Scopus	Springer Website
(TITLE-ABS-KEY ((“transfer learning” OR “knowledge transfer” OR “domain adaptation”) AND (mri OR “magnetic resonance” OR “diffusion imaging” OR “diffusion weighted imaging” OR “arterial spin labeling” OR “susceptibility mapping” OR bold OR “blood oxygenation level dependent” OR “blood oxygen level dependent”)))	(“transfer learning” OR “knowledge transfer” OR “domain adaptation”) AND (mri OR “magnetic resonance” OR “diffusion imaging” OR “diffusion weighted imaging” OR “arterial spin labeling” OR “susceptibility mapping” OR “T1” OR “T2”)

**Table 3 jimaging-07-00066-t003:** Tasks and applications in the surveyed papers.

Task (Total)	% of Studies	Application
Classification (68)	52.71%	Alzheimer’s diagnostics/prognostics (31), Tumor (10), fMRI decoding (6), Autism spectrum disorder (5), Injected cells (2), Parkinson (2), Schizophrenia (2), Sex (2), Aneurysm [27], Attention deficit hyperactivity disorder [28], Bipolar disorder [29], Embryonic neurodevelopmental disorders [30], Epilepsy [31], IDH mutation [32], Multiple sclerosis [33], Quality control [34]
Segmentation (45)	34.88%	Tumor (16), Anatomical (15), Lesion (14)
Regression (12)	9.30%	Age (8), Alzheimer’s disease progression [35], Autism symptom severity [36], Brain connectivity in Alzheimer’s disease [37], Tumor cell density [38]
Others (15)	11.63%	Reconstruction (5), Registration (4), Image translation (3), CBIR (2), Image fusion [39]

**Table 4 jimaging-07-00066-t004:** Transfer learning strategies categorization. Bold font highlights the proposed categories.

Type	% of Studies	Subtype	Subsubtype	Approaches
Instance (16)	11.63%	**Fixed**		6 (4.65%)
		**Optimized**	**Unsupervised**	5 (3.88%)
			**Supervised**	5 (3.88%)
Feature (38)	29.46%	Asymmetric		7 (5.43%)
		Symmetric	**Direct**	17 (13.18%)
			**Indirect**	14 (10.85%)
Parameter (87)	65.89%	Prior sharing		50 (38.76%)
		Parameter sharing	**One model**	21 (16.28%)
			**Multiple**	16 (12.40%)

## Data Availability

The data presented in this study are available in the Appendix A.

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
