# Peer review of "Transfer Learning in Magnetic Resonance Brain Imaging: A Systematic Review"

_2313-433X, 2021, doi:10.3390/jimaging7040066_

Round 1

Reviewer 1 Report

The Authors presented an article "Transfer Learning in Magnetic Resonance Brain Imaging: a Systematic Review". Journal of Imaging is an international, multi/interdisciplinary journal. Therefore, in many places I think should be some explanations. Also, there are some comments and questions for the work improvement. My suggestions you see below.

Specific comments:

1) You have classified transfer learning strategies following the criteria described in [9]. Can you explain if there are any other appropriate criteria in the other work, why you have chosen only these criteria? The classification is not so obvious to wide-range readers of Journal of Imaging readers. I think you can specify It in the first-mentioned place in the introduction and reveal it in more detail at the end of lines 98-106.

2) Is it possible to analyze the correlation between disease and the approach (Fig 2 and Fig 3 correlation)? Maybe It can be interesting.

3) In the conclusion part the more precise perspectives of the area should be inserted.

Minor comments

1) The lines 170-177 in the results section and Fig 1 maybe better present in the method section as one more subsection.

2) It will be better not to use the abbreviations in Fig 3. But it is in your consideration.

3) In the discussion section you have to present the links to Fig 2-3 and Tables 3 in par 5.1.

Reviewer 2 Report

This article is nicely written, covering most of the topics related to 'Transfer Learning in Magnetic Resonance Brain Imaging. 

Minor revision:

1) Adding a brief description about different magnetic resonance modality to the review will be helpful. 

2) Authors have not included Magnetic resonance spectroscopy in this study. Are there any transfer learning approached in this modality?

3) It is also useful to classify the articles based on modality (T1/T2/fmri/dwi/multi-model) and generate a figure similar to fig.2 and 3  

4) Add a short description about future research directions of Transfer Learning in Magnetic Resonance Brain Imaging.
